# Fast binary logistic regression

Nurdan Ayse Saran[1] and Fatih Nar[2]

[1] Department of Computer Engineering, Cankaya University, Ankara, Türkiye
[2] Department of Computer Engineering, Ankara Yildirim Beyazit University, Ankara, Türkiye

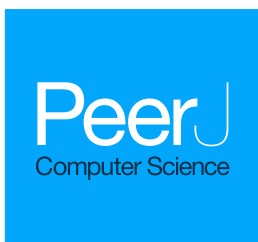

## ABSTRACT

This study presents a novel numerical approach that improves the training efficiency of binary logistic regression, a popular statistical model in the machine learning community. Our method achieves training times an order of magnitude faster than traditional logistic regression by employing a novel Soft-Plus approximation, which enables reformulation of logistic regression parameter estimation into matrix-vector form. We also adopt the $L_f$-norm penalty, which allows using fractional norms, including the $L_2$-norm, $L_1$-norm, and $L_0$-norm, to regularize the model parameters. We put $L_f$-norm formulation in matrix-vector form, providing flexibility to include or exclude penalization of the intercept term when applying regularization. Furthermore, to address the common problem of collinear features, we apply singular value decomposition (SVD), resulting in a low-rank representation commonly used to reduce computational complexity while preserving essential features and mitigating noise. Moreover, our approach incorporates a randomized SVD alongside a newly developed SVD with row reduction (SVD-RR) method, which aims to manage datasets with many rows and features efficiently. This computational efficiency is crucial in developing a generalized model that requires repeated training over various parameters to balance bias and variance. We also demonstrate the effectiveness of our fast binary logistic regression (FBLR) method on various datasets from the OpenML repository in addition to synthetic datasets.

## INTRODUCTION

Classification has been extensively investigated in statistical learning as a supervised learning task. As pioneering works, the contributions of *Berkson (1944)* and *Cox (1958)* to the field of logistic regression are recognized as foundational. Berkson applied the logistic function to bio-assay, while Cox introduced regression analysis of binary sequences. Different forms of logistic regression are available to handle outputs in binary, multinomial, and ordinal formats, making it a versatile machine learning algorithm capable of addressing a wide range of classification tasks such as marketing, finance, healthcare, and fraud detection. With the rapid development of machine learning, highly flexible models such as extreme gradient boosting (XGBoost), support vector machine (SVM), and neural networks have become prominent (*Cioffi et al., 2020*). However, studies indicating logistic regression's superior performance in some scenarios (*de Hond et al., 2022; Jiang, Hu & Jia, 2023; Nusinovici et al., 2020; Zabor et al., 2022*) and continuous usage of it (*Jaskie, Elkan & Spanias, 2019; Lu, 2024*) underscore the statistical machine learning community's continued interest in this method.

Corresponding author
Nurdan Ayse Saran,
buz@cankaya.edu.tr

Several improvements to the original logistic regression method are proposed (*Berkson, 1944*), such as weight regularization to obtain a generalized model or adapt the method for large data (*Jiang, Hu & Jia, 2023*; *Bertsimas, Pauphilet & Van Parys, 2021*; *Tibshirani, 1996*; *Zaidi et al., 2016*). Achieving a generalized model is closely tied to the bias-variance tradeoff where high-bias models risk oversimplification and underfitting, while low-bias models can overfit, exhibiting high variance. The tradeoff involves balancing these extremes to ensure the model generalizes well to unseen data without overfitting or underfitting. The optimal tradeoff is achieved by selecting appropriate model parameters that yield the most effective model complexity for training and validation sets, using techniques such as cross-validation and bootstrapping (*Wahba et al., 1998*; *Gong, 2006*; *Mohr & van Rijn, 2023*). It can be summarized that there must be a match between model complexity and data complexity to ensure that a model trained on training data performs well on unseen test data (*Geman, Bienenstock & Doursat, 1992*). Identifying optimal model parameters necessitates multiple training iterations with data subsets or random subsamples, a computationally intensive process for various parameter combinations, particularly with large datasets (*Mohr & van Rijn, 2023*; *Emmert-Streib & Dehmer, 2019*).

Logistic regression assumes a linear relationship between features and class distributions without demanding a specific statistical distribution. However, it is strictly required that features exhibit no multicollinearity. Regularization emerges as a vital solution to address multicollinearity and effectively manage model complexity. Aside from the intercept term, weight regularization applied to features encourages sparsity in the weight vector, resulting in fewer and more distinct features. This reduces multicollinearity and complexity, improving the model's generalization abilities (*Shi et al., 2010*; *Zhang et al., 2021*; *Avalos, Grandvalet & Ambroise, 2003*). Additionally, in scenarios with numerous features available but limited training data, the risk of overfitting can arise without an effective regularization technique (*Vapnik, 2006*). Thus, pursuing a sparse solution is crucial to reduce overfitting and prevent data's random pattern memorization (*Zhang et al., 2021*).

$L_2$-norm (also known as Ridge) regularization, first introduced by *Tikhonov (1963)*, penalizes the magnitudes of the weights to enforce smaller values, thereby increasing model robustness. In the literature, other weight regularization techniques (*Tikhonov & Arsenin, 1979*; *Morozov, 2012*; *Bertero, 1986*) also enforce sparsity for better robustness and increased model generalization. Among these techniques, $L_0$-norm[1], provide most sparse solutions (*Wang, Chen & Yang, 2022*; *Greenwood et al., 2020*). Nevertheless, using $L_0$-norm is challenging since it is not convex. Thereby, $L_1$-norm, namely least absolute shrinkage and selection operator (LASSO) (*Tibshirani, 1996*), is proposed as a convex relaxation of non-convex $L_0$-norm. Each regularization approach imposes different constraints on the model training. For example, $L_2$-norm imposes a squared penalty (*Hoerl & Kennard, 1970*) and $L_1$-norm imposes an absolute penalty on the model's parameters (*Ozgur, Nar & Erdem, 2018*; *Wei et al., 2019*; *Xie et al., 2023*) while $L_0$-norm imposes a constant penalty for all non-zero weights (*Wang, Chen & Yang, 2022*; *Greenwood et al., 2020*). Note that the Bayesian information criterion (BIC) (*Schwarz, 1978*) and the Akaike information criterion (AIC) (*Akaike, 1998*), well-known model selection criteria, are

---

[1] $L_0$ is not a norm but a pseudo-norm since it does not satisfy all norm axioms

special cases of $L_0$-norm regularization. So, sparse models using $L_0$-norm and $L_1$-norm enforce many coefficients to shrink to exactly zero while $L_2$-norm tends to produce model parameters closer but not precisely zero (*Pereyra et al., 2017*). Thus, $L_1$-norm and $L_0$-norm are employed in feature selection to avoid overfitting. On the contrary, $L_2$-norm has better computational efficiency than $L_1$-norm and is even more efficient than $L_0$-norm. Thereby, several studies prefer to use $L_1$-norm or $L_2$-norm regularized logistic regression for large scale data (*Koh, Kim & Boyd, 2007*; *Shi et al., 2010*; *Jovanovich & Lazar, 2012*; *Su, 2020*). Elastic-net combines the $L_1$-norm and $L_2$-norm penalties, so it tends to choose more features than LASSO, but computational efficiency becomes similar to Ridge. Recently, some studies tackled the $L_0$-norm regularization for logistic regression while dealing with challenges of using non-convex $L_0$-norm regularization term (*Ming & Yang, 2024*; *Hazimeh, Mazumder & Nonet, 2023*; *Knauer & Rodner, 2023*; *Deza & Atamturk, 2022*). Although not proposed for logistic regression, the use of fractional norm as a penalty term was proposed as a flexible and practical approach for enforcing smoothness on the image denoising (*Ozcan, Sen & Nar, 2016*), where it is also better suited for handling the generalized Gaussian distribution of the variables (*Bernigaud et al., 2021*).

As already stated in the literature, the maximum likelihood estimation of the logistic regression has some shortcomings:

- It may struggle to handle massive sparse datasets (*Holland & Welsch, 1977*; *Li, Zhu & Wang, 2023*) effectively.
- Logistic regression often struggles with imbalanced data, tending to favor the majority class and resulting in poor performance on the minority class (*King & Zeng, 2001*).
- For the training set where the number of samples is smaller than the number of features, directly solving the logistic regression is an ill-posed problem (*Liu, Chen & Ye, 2009*).
- It is sensitive to anomalous data and collinearity (*Feng et al., 2014*; *Midi, Sarkar & Rana, 2010*).
- Oversampling data instances may decrease estimation performance and increase computational expenses (*Wang, 2020*).

One strategy to mitigate these challenges involves utilizing the iteratively reweighted least squares method with an appropriate solver to construct binary logistic regression classifiers, particularly for large-scale datasets (*Paciorek, 2007*; *Rouhani-Kalleh, 2007*). Employing appropriate numerical solvers can also address these problems. Hence, numerous numerical solvers have been deployed to tackle these challenges, such as gradient descent and its variants, Newton's method, and quasi-Newton methods. These solvers play a crucial role in training models efficiently by iteratively adjusting parameters to minimize a specified objective function. Although these approaches attempt to eliminate anomalies or collinearity problems in the data, working with many solvers with different regularizers poses a challenging issue for researchers (*Liu, Chen & Ye, 2009*). The selection of the solver typically depends on the specific characteristics of the data and the nature of the problem under consideration.

Scikit-learn (*Pedregosa et al., 2011*), a versatile Python library integrating a wide range of state-of-the-art machine learning algorithms, has become one of the most popular choices among many libraries due to its simplicity, comprehensive documentation, and extensive community support. The LogisticRegression class in scikit-learn already incorporates several optimization methods (see Table 1), each tailored for different regularizers, data sizes, and computational needs. Similarly, other open-source libraries, including cuML and Statsmodels, provide their own implementations of logistic regression, each with a variety of solvers and parameter configurations to enhance flexibility and efficiency. In particular, cuML offers graphics processing unit (GPU)-accelerated solvers for large-scale computations and Statsmodels provides robust statistical modeling options for binary response variables.

With increasing dataset sizes, logistic regression needs to improve in terms of efficiency and scalability. In response, advanced versions have emerged, designed to prioritize speed and accuracy. These innovative approaches accelerate learning by leveraging techniques such as dimensionality reduction, parallel processing, and advanced data structures.

Logistic regression training is formulated to solve an optimization problem based on likelihood maximization. However, calculating the gradient with all training data in each iteration for large datasets becomes computationally demanding. Therefore, *Song et al. (2021)* proposed an adaptive sampling method in which the gradient estimation is divided into several sub-problems according to the data size, and then each sub-problem is solved independently before combining the results of all sub-problems. *Shi et al. (2010)* suggested a hybrid algorithm that merges two optimization iterations: one that is fast and memory-efficient and another that is slower but yields more precise results.

Various approaches have been proposed to enhance the speed of logistic regression, which can generally be categorized into efficient numerical methods and software-based improvements. Table 1 summarizes the solvers employed in scikit-learn. A notable earlier study by *Komarek & Moore (2003)* utilizes Cholesky decomposition to accelerate logistic regression. However, since many contemporary methods now incorporate Cholesky decomposition within NumPy or employ more advanced solvers, this study has yet to be considered state-of-the-art. Additionally, some approaches leverage field programmable gate array (FPGA) and GPU (*Wienbrandt et al., 2019*), stochastic gradient descent, and mini-batch (*Yang et al., 2019*; *Liang et al., 2020*; *Jurafsky & Martin, 2024*) methods to improve the speed of logistic regression. These techniques are often complementary and can be integrated with other methods, representing practical efforts to accelerate logistic regression further.

We introduce a novel numerical approach that remarkably improves the training efficiency of binary logistic regression without relying on dimension reduction or parallel processing and without requiring feature independence. This efficiency is obtained by employing a novel Soft-Plus approximation, which enables reformulation of logistic regression parameter estimation into matrix-vector form. Additionally, unlike the multiple solvers employed in scikit-learn's logistic regression for different regularization schemas, we present a single flexible $L_f$-norm regularization approach that also provides flexibility to include or exclude penalization of the intercept term. Our regularization approach

**Table 1 Minimizers employed as solvers in scikit-learn's logistic regression algorithm.**

|  | $L_1$ | $L_2$ | ElasticNet | No regularization |
|---|:---:|:---:|:---:|:---:|
| LBFGS[1] |  | ✓ |  | ✓ |
| LIBLINEAR (*Fan et al., 2008*) | ✓ | ✓ |  |  |
| Newton-CG[1] |  | ✓ |  | ✓ |
| Newton-Cholesky[1] |  | ✓ |  | ✓ |
| SAG (*Schmidt, Roux & Bach, 2017*) |  | ✓ |  | ✓ |
| SAGA (*Defazio, Bach & Lacoste-Julien, 2014*) | ✓ | ✓ | ✓ | ✓ |

**Note:**
[1] See *Nocedal & Wright (2006)* for further information on these common optimization approaches.

supports a range of weight penalties through a unified codebase with a specifically designed numerical minimizer. To demonstrate the computational efficiency of our method, we conducted several quantitative experiments, comparing our method against the widely used scikit-learn library. Here, we used benchmark data from OpenML, an open machine learning data repository, and synthetic data designed for controlled experiments. Experiments demonstrate that our method effectively handles collinear features and large data while providing superior efficiency with minimal to no loss in accuracy. Our fast binary logistic regression (FBLR) exclusively utilizes the Python NumPy library. As a result, it benefits from all current capabilities of NumPy and upcoming enhancements while keeping the codebase simple. This streamlined yet powerful approach allows it to surpass the performance of scikit-learn's logistic regression (LR) implementation in processing speed, which relies on well-established and highly optimized libraries. Our experiments demonstrate that FBLR achieves an average speedup of an order of magnitude faster, with the maximum observed speedup reaching 48.3 times.

# PROPOSED METHOD

## Fast binary logistic regression

For logistic regression, the likelihood function in the context of maximum likelihood estimation (MLE) is formulated based on the assumption that each observation is an independent Bernoulli trial. Consider a dataset with $n$ observations $(x_i, y_i)$, where each $x_i \in \mathbb{R}^d$ pairs with a binary outcome $y_i \in \{0, 1\}$. Here, $i$ is the data index, and $d$ is the number of features, including the intercept. Then, the logistic regression model predicts the probability of the outcome being 1 is defined as:

$$P(y_i = 1 | x_i) = \frac{e^{x_i^\top w}}{1 + e^{x_i^\top w}} \tag{1}$$

where $w \in \mathbb{R}^d$ represents the model parameters, namely the weight vector and $x_i = [1 \ x_{i,1} \ x_{i,2} \ \cdots \ x_{i,d-1}]^\top$ is the $i^{\text{th}}$ data vector. For the binary classification using logistic regression, likelihood $l(w)$ of observing the entire dataset given $w$ is the product of the probabilities for each observation that is given in Eq. (2):

$$l(w) = \prod_i P(y_i = 1 | x_i) \prod_i P(y_i = 0 | x_i) = \prod_{i:y_i=1} P(x_i) \prod_{i:y_i=0} (1 - P(x_i)) \tag{2}$$

The optimum model parameter is found by maximizing the $l(w)$ with respect to weight vector $w$ for the given input data and corresponding target class labels, known as MLE:

$$\text{argmax}_w \; l(w) \tag{3}$$

Optimizing $l(w)$ presents challenges due to the presence of product terms. However, instead of maximizing $l(w)$, one can opt to minimize $-\log l(w)$. This approach is feasible because the maximum of $l(w)$ corresponds to the minimum of $-\log l(w)$, and $-\log l(w)$ is significantly easier to optimize. Then, we simplify $-\log l(w)$ as below:

$$
\begin{aligned}
-\log l(w) &= -\left[ \sum_{i:y_i=1} \log P(x_i) + \sum_{i:y_i=0} \log(1 - P(x_i)) \right] \\
&= \sum_{i:y_i=0} \log(1 + e^{x_i^\top w}) - \sum_{i:y_i=1} \left( x_i^\top w - \log(1 + e^{x_i^\top w}) \right) \\
&= \sum_i \left( \log(1 + e^{x_i^\top w}) - y_i x_i^\top w \right).
\end{aligned}
\tag{4}
$$

To develop an efficient minimizer for $-\log l(w)$, we approximate the Soft-Plus function, $\log(1 + e^{x_i^\top w})$, at $\hat{w}$ to obtain a quadratic form where $\hat{w}$ is a proxy constant for $w$. Details of our novel quadratic Soft-Plus approximation are provided in "Approximation of Soft-Plus". By substituting this Soft-Plus approximation into Eq. (4), we derive the following form of $-\log l(w)$:

$$-\log l(w) \approx \sum_i \left( z_i (x_i^\top w)^2 + \left( \frac{1}{2} - y_i \right)(x_i^\top w) + \log 2 \right) \tag{5}$$

where

$$
z_i = \begin{cases}
\frac{1}{8}, & x_i^\top \hat{w} = 0 \\
\frac{\log(1 + e^{x_i^\top \hat{w}}) - \log(2)}{(x_i^\top \hat{w})^2} - \frac{1}{2 x_i^\top \hat{w}}, & \text{otherwise.}
\end{cases}
\tag{6}
$$

Finally, we put Eq. (5) into matrix-vector form as below:

$$
\begin{aligned}
-\log l(w) &\approx (\mathbf{X}w)^\top \mathbf{Z}(\mathbf{X}w) + \left( \frac{\vec{1}}{2} - y \right)^\top \mathbf{X}w + n\log 2 \\
&\approx w^\top \mathbf{X}^\top \mathbf{Z}\mathbf{X}w + \left( \frac{\vec{1}}{2} - y \right)^\top \mathbf{X}w + n\log 2
\end{aligned}
\tag{7}
$$

where $\mathbf{X} \in \mathbb{R}^{n \times d}$ is input data in matrix form, $y \in \mathbb{R}^{n \times 1}$ is target data in vector form, $w \in \mathbb{R}^{d \times 1}$ is weight vector, $\mathbf{Z} \in \mathbb{R}^{n \times n}$ is a diagonal matrix form of $z_i$.

Various forms of weight regularization have been proposed in the literature for logistic regression, each requiring distinct numerical approaches. Alternatively, we use smooth $L_f$-norm (*Ozcan, Sen & Nar, 2016*) regularization to achieve a unified framework that only requires a single numerical approach, incorporating Ridge, pseudo ElasticNet, LASSO, $L_0$-norm (notably, a pseudo-norm), and other fractional norm regularizations.

For quadratic approximation, let $\hat{\rho}$ be a constant proxy for $\rho$ and define $u$ as a constant coefficient, $u = \left(|\hat{\rho}|^{2-f} + \varepsilon\right)^{-1}$. Then, quadratically approximated $L_f$-norm is

$$L_f(\rho) \approx \rho^2(|\hat{\rho}|^{2-f} + \varepsilon)^{-1} = u\rho^2. \tag{8}$$

Refer to "Approximation of $L_f$-Norm" for details and justification of the quadratic $L_f$-norm approximation.

An iterative minimization approach (solver) is now required as we utilize quadratic approximations of $\log(1 + e^{x_i^\top w})$ (Soft-Plus) and a smooth $L_f$-norm. Both approximations are performed on $w$ for each iteration, where the applied approximations are accurate near the point of approximation, namely $\hat{w}$ which is a proxy constant for $w$. To ensure that the new solution remains close to the approximation point at the current iteration, we introduce a slow-step-regularization (SSR) term with a coefficient $\lambda$. Finally, we normalize the terms by the data count and weight vector dimension.

$$J^{(k)}(w) = -\frac{1}{n}\log l(w) + \frac{\lambda}{2d}(w - \hat{w})^\top(w - \hat{w}) + \frac{\gamma}{2d}p^\top L_f(w) \tag{9}$$

where $k$ is the iteration index, $w$ is weight vector, $\hat{w}$ is a proxy constant for $w$ at $k^{\text{th}}$ iteration, $\lambda$ is the SSR coefficient, and $\gamma$ is $L_f$-norm regularization coefficient. Here, $p$ is the vector used to prevent penalizing the intercept term, defined as $p = [\ldots, p_j, \ldots]$, where $p_j$ is 0 for the intercept (to avoid penalizing it) and 1 for all other coefficients. Then, $p^\top L_f(w)$ is approximated as:

$$p^\top L_f(w) = \sum_{j=1}^{d} p_j L_f(w_j) = \sum_{j=1}^{d} h_j w_j^2 = w^\top \mathbf{H} w \tag{10}$$

where $\mathbf{H} \in \mathbb{R}^{d \times d}$ is a diagonal matrix form of $h_j$ values that is defined as below ($\varepsilon$ is a small positive constant with default as $\varepsilon = 10^{-10}$):

$$h_j = p_j\left(|\hat{w}_j|^{2-f} + \varepsilon\right)^{-1}. \tag{11}$$

The obtained cost function $J^{(k)}(w)$, comprising only quadratic terms, linear terms, and constants, can be expressed in a matrix-vector form.

$$J^{(k)}(w) = \frac{1}{n}w^\top \mathbf{X}^\top \mathbf{Z} \mathbf{X} w + \frac{1}{n}\left(\frac{\vec{1}}{2} - y\right)^\top \mathbf{X} w + \log 2 + \frac{\lambda}{2d}(w - \hat{w})^\top(w - \hat{w}) + \frac{\gamma}{2d}w^\top \mathbf{H} w. \tag{12}$$

Afterwards, we can minimize $J^{(k)}(w)$ with respect to $w$ at $k^{\text{th}}$ iteration by taking the derivative of $J^{(k)}(w)$ with respect to $w$ and equalizing it to zero:

$$\frac{\partial J^{(k)}(w)}{\partial w} = \frac{1}{n}2\mathbf{X}^\top \mathbf{Z} \mathbf{X} w + \frac{1}{n}\mathbf{X}^\top\left(\frac{\vec{1}}{2} - y\right) + \frac{\lambda}{2d}2(w - \hat{w}) + \frac{\gamma}{2d}2\mathbf{H}w = 0 \tag{13}$$

which can be arranged as:

$$\left(\frac{2}{n}\mathbf{X}^\top \mathbf{Z}\mathbf{X} + \frac{\lambda}{d}\mathbf{I} + \frac{\gamma}{d}\mathbf{H}\right)w = \frac{\lambda}{d}\hat{w} + \frac{1}{n}\mathbf{X}^\top\left(y - \frac{\vec{1}}{2}\right) \tag{14}$$

where $\mathbf{I} \in \mathbb{R}^{d \times d}$ is the identity matrix. Note that Eq. (14) can be represented as a linear system $\mathbf{A}w^{(k+1)} = b$. Finally, the solution can be represented as below where $\mathbf{Z}$ and $\mathbf{H}$ are updated in each iteration. In our case, $\mathbf{A}$ is a symmetric matrix, and $v$ is a constant vector that can be computed before the iterative minimization.

| Initialization | Iteration |
|---|---|
| $v = \frac{1}{n}\mathbf{X}^\top(y - \frac{\vec{1}}{2})$ | $\mathbf{A}w^{(k+1)} = b$ |
| | $\mathbf{A} = \frac{2}{n}\mathbf{X}^\top\mathbf{Z}\mathbf{X} + \frac{\lambda}{d}\mathbf{I} + \frac{\gamma}{d}\mathbf{H}$ |
| | $b = \frac{\lambda}{d}w^{(k)} + v$ |

Data matrix $\mathbf{X}$ can be rank-deficient due to colinear features or insufficient data samples. When the data matrix $\mathbf{X}$ is rank-deficient, and no regularization technique is applied ($\lambda = 0$), the matrix $\mathbf{A}$ may become a symmetric positive semi-definite matrix. Nonetheless, if $\mathbf{X}$ is of full rank or a regularization is applied ($\lambda > 0$), the matrix $\mathbf{A}$ becomes symmetric positive definite. To mitigate the challenges due to rank-deficiency and the subsequent numerical instabilities, we utilized low-rank approximation using singular value decomposition (SVD) (*Lawson & Hanson, 1995*; *Hansen, 1990*) for approximating the data matrix $\mathbf{X} \in \mathbb{R}^{n \times d}$ (*Ye, 2005*).

$$\mathbf{U},\mathbf{S},\mathbf{V}^\top = \text{Truncated-SVD}(\mathbf{X}) \tag{15}$$

where $\mathbf{X} = \mathbf{U}\mathbf{S}\mathbf{V}^\top$, $\mathbf{U} \in \mathbb{R}^{n \times d}$, $\mathbf{S} \in \mathbb{R}^{d \times d}$, and $\mathbf{V} \in \mathbb{R}^{d \times d}$. Then

$$\begin{aligned}
\mathbf{A} &= \frac{2}{n}(\mathbf{U}\mathbf{S}\mathbf{V}^\top)^\top \mathbf{Z}(\mathbf{U}\mathbf{S}\mathbf{V}^\top) + \frac{\lambda}{d}\mathbf{I} + \frac{\gamma}{d}\mathbf{H} = \frac{2}{n}\mathbf{V}\mathbf{S}^\top\mathbf{U}^\top\mathbf{Z}\mathbf{U}\mathbf{S}\mathbf{V}^\top + \frac{\lambda}{d}\mathbf{I} + \frac{\gamma}{d}\mathbf{H} \\
&= \frac{2}{n}\mathbf{V}\mathbf{S}(\mathbf{U}^\top\mathbf{Z}\mathbf{U})\mathbf{S}\mathbf{V}^\top + \frac{\lambda}{d}\mathbf{I} + \frac{\gamma}{d}\mathbf{H}.
\end{aligned} \tag{16}$$

We use $r$-rank approximation of $\mathbf{U}$, $\mathbf{S}$, and $\mathbf{V}$ matrices such that $\mathbf{X} \approx \widehat{\mathbf{U}}\widehat{\mathbf{S}}\widehat{\mathbf{V}}^\top$. Note that, $\widehat{\mathbf{U}} \in \mathbb{R}^{n \times r}$, $\widehat{\mathbf{S}} \in \mathbb{R}^{r \times r}$, and $\widehat{\mathbf{V}} \in \mathbb{R}^{d \times r}$ denotes low-rank approximations of the $\mathbf{U}$, $\mathbf{S}$, and $\mathbf{V}$ matrices, respectively. Here, $\widehat{\mathbf{U}}$ and $\widehat{\mathbf{V}}$ are orthonormal matrices and $\widehat{\mathbf{S}}$ is a diagonal matrix. In this low-rank approximation, $r$ is the rank such that $r \leq d$ and $r$ is chosen such that almost all of the energy (by eigenvalues) is preserved.

We initialize $w^{(0)}$ using least-square to start from a reasonable initial point. For computational efficiency, we also use the SVD decomposition $\mathbf{X} = \mathbf{U}\mathbf{S}\mathbf{V}^\top$ on the least-square equation:

$$w^{(0)} = (\mathbf{X}^\top\mathbf{X})^{-1}\mathbf{X}^\top y = (\mathbf{V}\mathbf{S}^\top\mathbf{U}^\top\mathbf{U}\mathbf{S}\mathbf{V}^\top)^{-1}\mathbf{V}\mathbf{S}^\top\mathbf{U}^\top y = \mathbf{V}\mathbf{S}^{-1}\mathbf{U}^\top y \tag{17}$$

To circumvent numerical challenges and gain further computational efficiency, we employ a low-rank approximation as $w^{(0)} = \widehat{\mathbf{V}}\widehat{\mathbf{S}}^{-1}\widehat{\mathbf{U}}^\top y = \mathbf{F}y$ where $\mathbf{F} = \widehat{\mathbf{V}}\widehat{\mathbf{S}}^{-1}\widehat{\mathbf{U}}^\top$. Computing $\mathbf{F}$ is computationally efficient since the inverse of the diagonal matrix $\widehat{\mathbf{S}}$ is

computationally cheap. Also, computing $w^{(0)} = \mathbf{F}y$ is cheap as well. For both with and without regularization, a common initialization is defined as follows:

---

$\mathbf{U}, \mathbf{S}, \mathbf{V}^\top = \text{Truncated-SVD}(\mathbf{X})$ since $\mathbf{X} = \mathbf{USV}^\top$

Determine rank $r$ such that $\mathbf{X} \approx \widehat{\mathbf{U}}\widehat{\mathbf{S}}\widehat{\mathbf{V}}^\top$

$\widehat{\mathbf{U}} = r\text{-rank}(\mathbf{U})$ and $\widehat{\mathbf{S}} = r\text{-rank}(\mathbf{S})$ and $\widehat{\mathbf{V}} = r\text{-rank}(\mathbf{V})$

$w^{(0)} = \mathbf{F}y$ where $\mathbf{F} = \widehat{\mathbf{V}}\widehat{\mathbf{S}}^{-1}\widehat{\mathbf{U}}^\top$

---

There are two paths for the developed iterative minimization: (a) with regularization and (b) without regularization. It should be noted that applying $L_f$-norm regularization requires setting a positive $\gamma$ value, which in turn necessitates assigning a positive $\lambda$ parameter to enable SSR regularization.

**a)** With regularization ($\lambda > 0$ and $\gamma > 0$)

Recall, first we need to construct $\mathbf{A}$ matrix and $b$ vector then solve the linear system $\mathbf{A}w^{(k+1)} = b$. Using the $v = \frac{1}{n}\mathbf{X}^\top(y - \frac{\vec{1}}{2})$ we have:

$$\mathbf{A} = \frac{2}{n}\mathbf{X}^\top\mathbf{Z}\mathbf{X} + \frac{\lambda}{d}\mathbf{I} + \frac{\gamma}{d}\mathbf{H} \text{ and } \mathbf{b} = \frac{\lambda}{d}w^{(k)} + v. \tag{18}$$

With $\mathbf{G} = \widehat{\mathbf{S}}\widehat{\mathbf{V}}^\top$, low-rank approximation of $\mathbf{X} \approx \widehat{\mathbf{U}}\widehat{\mathbf{S}}\widehat{\mathbf{V}}^\top$ leads to $\mathbf{X} \approx \widehat{\mathbf{U}}\mathbf{G}$:

$$\begin{aligned}
\mathbf{A} &= \frac{2}{n}\mathbf{X}^\top\mathbf{Z}\mathbf{X} + \frac{\lambda}{d}\mathbf{I} + \frac{\gamma}{d}\mathbf{H} = \frac{2}{n}(\widehat{\mathbf{U}}\widehat{\mathbf{S}}\widehat{\mathbf{V}}^\top)^\top\mathbf{Z}(\widehat{\mathbf{U}}\widehat{\mathbf{S}}\widehat{\mathbf{V}}^\top) + \frac{\lambda}{d}\mathbf{I} + \frac{\gamma}{d}\mathbf{H} \\
&= \frac{2}{n}\mathbf{G}^\top(\widehat{\mathbf{U}}^\top\mathbf{Z}\widehat{\mathbf{U}})\mathbf{G} + \frac{\lambda}{d}\mathbf{I} + \frac{\gamma}{d}\mathbf{H}.
\end{aligned} \tag{19}$$

Let compute $v$ in order to compute $b$ and use $r$-rank approximation of $v$ as below:

$$v = \frac{1}{n}\mathbf{X}^\top\left(y - \frac{\vec{1}}{2}\right) = \frac{1}{n}\mathbf{G}^\top\left(\widehat{\mathbf{U}}^\top\left(y - \frac{\vec{1}}{2}\right)\right). \tag{20}$$

Finally, the numerical minimization approach with regularization is as follows:

| Initialization | Iteration |
|---|---|
| $\mathbf{G} = \widehat{\mathbf{S}}\widehat{\mathbf{V}}^\top$ where $\mathbf{X} \approx \widehat{\mathbf{U}}\widehat{\mathbf{S}}\widehat{\mathbf{V}}^\top$ | $\mathbf{A}w^{(k+1)} = b$ |
| with low-rank estimation using SVD | $\mathbf{A} = \frac{2}{n}\mathbf{G}^\top(\widehat{\mathbf{U}}^\top\mathbf{Z}\widehat{\mathbf{U}})\mathbf{G} + \frac{\lambda}{d}\mathbf{I} + \frac{\gamma}{d}\mathbf{H}$ |
| $v = \frac{1}{n}\mathbf{G}^\top(\widehat{\mathbf{U}}^\top(y - \frac{\vec{1}}{2}))$ | $b = \frac{\lambda}{d}w^{(k)} + v$ |

Thanks to the low-rank approximation applied to the data matrix $\mathbf{X}$ in addition to applied $L_f$-norm and SSR regularization, the proposed cost function becomes strictly convex. So, the linear system $\mathbf{A}w^{(k+1)} = b$ is well-conditioned, ensuring a unique solution for $w^{(k+1)}$, whereas the logistic regression cost function remains only convex due to potential rank deficiencies in $\mathbf{X}$.

**b)** Without regularization ($\lambda = 0$ and $\gamma = 0$)

First $\mathbf{A}$ and $b$ are constructed then linear system $\mathbf{A}w^{(k+1)} = b$ is solved.

$$\frac{2}{n}\left(\mathbf{X}^\top \mathbf{Z} \mathbf{X}\right)w^{(k+1)} = \frac{1}{n}\mathbf{X}^\top\left(y - \frac{\vec{1}}{2}\right)$$

$$w^{(k+1)} = \frac{1}{2}\left(\mathbf{X}^\top \mathbf{Z} \mathbf{X}\right)^{-1}\mathbf{X}^\top\left(y - \frac{\vec{1}}{2}\right). \tag{21}$$

Let us use low-rank approximation of $\mathbf{X} \approx \widehat{\mathbf{U}}\widehat{\mathbf{S}}\widehat{\mathbf{V}}^\top$:

$$w^{(k+1)} \approx \frac{1}{2}\left((\widehat{\mathbf{U}}\widehat{\mathbf{S}}\widehat{\mathbf{V}}^\top)^\top \mathbf{Z}(\widehat{\mathbf{U}}\widehat{\mathbf{S}}\widehat{\mathbf{V}}^\top)\right)^{-1}(\widehat{\mathbf{U}}\widehat{\mathbf{S}}\widehat{\mathbf{V}}^\top)^\top\left(y - \frac{\vec{1}}{2}\right)$$

$$\approx \frac{1}{2}\widehat{\mathbf{V}}\widehat{\mathbf{S}}^{-1}\widehat{\mathbf{U}}^\top \mathbf{Z}^{-1}\widehat{\mathbf{U}}\widehat{\mathbf{U}}^\top\left(y - \frac{\vec{1}}{2}\right) \tag{22}$$

$$\approx \frac{1}{2}\mathbf{F}\mathbf{Z}^{-1}y_q \text{ where } y_q = \widehat{\mathbf{U}}\left(\widehat{\mathbf{U}}^\top\left(y - \frac{\vec{1}}{2}\right)\right).$$

The final iterative numerical minimization approach without regularization is as follows:

| Initialization | Iteration |
|---|---|
| $y_q = \widehat{\mathbf{U}}(\widehat{\mathbf{U}}^\top(y - \frac{\vec{1}}{2}))$ | $w^{(k+1)} \approx \frac{1}{2}\mathbf{F}\mathbf{Z}^{-1}y_q$ |

Note that $\mathbf{Z}$ is a diagonal matrix with all positive entries, making its inverse extremely efficient to compute. Additionally, $\mathbf{F}\mathbf{Z}^{-1}y_q$ can be calculated efficiently, as computing $\mathbf{Z}^{-1}y_q$ requires only $n$ operations, and multiplying $\mathbf{F}$ by the resulting vector requires only $d \times n$ operations. As a result, each iteration without regularization remains highly efficient. In addition to computational efficiency, it is guaranteed that $\mathbf{F}\mathbf{Z}^{-1}$ is always well-conditioned, and a unique solution for $w^{(k+1)}$ exists since $\mathbf{Z}$, as a diagonal matrix with positive entries, is always invertible. Thus, the employed low-rank approximation regularizes the solution to mitigate possible collinearity in the data matrix $\mathbf{X}$.

## Implementation

This section outlines the implementation of the proposed method, with the Low-Rank approximation described in the Algorithm 1 and the proposed FBLR method presented in the Algorithm 2. In the Algorithm 1, $\mathbf{X}$ is input data matrix, $\xi$ is energy-percentile (default is 99.9999) the DIMENSION function returns the number of rows ($n$) and the number of features ($d$). The SVD$^{(*)}$ function carries out singular value decomposition, choosing the most suitable variant—either truncated or randomized SVD (refer to "Randomized SVD"), or SVD with row reduction (SVD-RR) (refer to "SVD with Row Reduction")—based on an evaluation of $n$ and $d$. Lastly, the RANK function (see "Determining the Rank") in the Algorithm 1 finds the matrix's rank by analyzing the eigenvalues derived from $\mathbf{S}$, consequently generating matrices of low rank.

---

**Algorithm 1** Procedure to obtain a low-rank approximation of the data matrix X.

1: **procedure** LowRankApproximation $(\mathbf{X}, \xi)$

2:     *Returns the dimension of the data matrix* $\mathbf{X}$

3:     $n, d \leftarrow$ DIMENSION$(\mathbf{X})$        $\triangleright$ $n$: #row, $d$: #feature

4:     *Utilizing either Truncated SVD or Randomized SVD*

5:     $\mathbf{U}, \mathbf{S}, \mathbf{V}^\top \leftarrow \text{SVD}^{(*)}(\mathbf{X})$        $\triangleright$ $n$ see "SVD with Row Reduction (SVD-RR) & Randomized SVD"

6:     *Computes the rank of matrix* $\mathbf{X}$ *using eigenvalues*

7:     $r \leftarrow$ RANK$(\mathbf{S}, \xi)$        $\triangleright$ $n$ see "Determining the Rank"

8:     *The low-rank components of the matrices are extracted*

9:     $\widehat{\mathbf{U}} \leftarrow \mathbf{U}[:, 1:r]$

10:     $\widehat{\mathbf{S}} \leftarrow \mathbf{S}[1:r, 1:r]$

11:     $\widehat{\mathbf{V}} \leftarrow \mathbf{V}[1:r, 1:r]$

12:     **return** $n, d, r, \widehat{\mathbf{U}}, \widehat{\mathbf{S}}, \widehat{\mathbf{V}}$

13: **end procedure**

---

The pseudocode for the proposed FBLR method is given in Algorithm 2. In this algorithm, operations that remain constant throughout iterations are performed upfront in the initialization phase. First, low-rank approximation of the matrix $\mathbf{X}$ is obtained using Algorithm 1. Depending on whether regularization is applied, various matrices and vectors are precomputed accordingly. Subsequently, at each iteration, a dense linear system is constructed and solved using the Cholesky decomposition, leveraging the fact that $\mathbf{A}$ is a symmetric positive definite matrix when regularization is employed (line 20 in Algorithm 2). Even when regularization is not employed, computing $w$ is still well-conditioned (line 22 in Algorithm 2) since $\mathbf{Z}$ is a diagonal matrix with all positive entries.

Using Python v3.12.1, NumPy v1.26.4, and scikit-learn v1.3.2, proposed FBLR method in Algorithm 2, is implemented as the Python class `FastLogisticRegressionLowRank` extending scikit-learn's abstract `BaseEstimator` and `ClassifierMixin` classes. We use NumPy for all matrix and vector operations where our NumPy configuration uses OpenBLAS as the backend for basic linear algebra subprograms (BLAS) operations in our environment. For additional efficiency, NumPy also utilizes CPU instructions such as single instruction multiple data (SIMD).

## Computational complexity analysis

In the Algorithm 2, the time complexity of the initialization phase is $O(nd^2)$ since the time complexity of both the SVD used in the low-rank approximation and $\mathbf{F} = \widehat{\mathbf{V}}\widehat{\mathbf{S}}^{-1}\widehat{\mathbf{U}}^\top$ is $O(nd^2)$. At the same time, the remaining operations have lower time complexity. Furthermore, the computational complexity per iteration is $O(nd^2)$ when incorporating regularization ($L_f$-norm & SSR) and $O(nd)$ without regularization. Noting that $k$ represents the number of iterations executed until the Algorithm 2 converges, the total time complexity of the FBLR method is $O(nd^2k)$ for cases with regularization and

---

**Algorithm 2 Fast binary logistic regression (FBLR).**

1: **procedure** FBLR($\mathbf{X}$, $y$, $f$, $\lambda$, $\gamma$, $\xi$, $K$, $C_{tolerance}$)

2:    Initialization

3:    $n, d, r, \widehat{\mathbf{U}}, \widehat{\mathbf{S}}, \widehat{\mathbf{V}}^{\top} \leftarrow \text{LowRankApproximation}(\mathbf{X}, \xi)$

4:    $\mathbf{F} = \widehat{\mathbf{V}}\widehat{\mathbf{S}}^{-1}\widehat{\mathbf{U}}^{\top}$

5:    $w = \mathbf{F}y$               $\triangleright$ $n$ Initializing weights *via* the pseudo-inverse

6:    **if** ($\lambda > 0$ or $\gamma > 0$) **then**

7:       $\mathbf{G} \leftarrow \widehat{\mathbf{S}}\widehat{\mathbf{V}}^{\top}$

8:       $v \leftarrow \frac{1}{n}\mathbf{G}^{\top}\left(\widehat{\mathbf{U}}^{\top}(y - \frac{\vec{1}}{2})\right)$

9:    **else**

10:      $y_q \leftarrow \widehat{\mathbf{U}}\left(\widehat{\mathbf{U}}^{\top}(y - \frac{\vec{1}}{2})\right)$

11:   **end if**

12:   Iteration

13:   **for** $k \leftarrow 0$ **to** $K$ **do**

14:      $\widehat{w} \leftarrow w$

15:      $\mathbf{Z} \leftarrow \text{diag}(z)$               $\triangleright$ $z = \text{vector}(z_i)$, Eq. (6)

16:      **if** ($\lambda > 0$ or $\gamma > 0$) **then**

17:         $\mathbf{H} \leftarrow \text{diag}(h)$            $\triangleright$ $h = \text{vector}(h_i)$, Eq. (11)

18:         $\mathbf{A} \leftarrow \frac{2}{n}\mathbf{G}^{\top}\left(\widehat{\mathbf{U}}^{\top}\mathbf{Z}\widehat{\mathbf{U}}\right)\mathbf{G} + \frac{\lambda}{d}\mathbf{I} + \frac{\gamma}{d}\mathbf{H}$

19:         $b \leftarrow \frac{\lambda}{d}w^{(k)} + v$

20:         $w \leftarrow \text{solve}(\mathbf{A}, b)$        $\triangleright$ $h$ Applying Cholesky decomposition for solving $Ax = b$

21:      **else**

22:         $w \leftarrow \frac{1}{2}\mathbf{F}\mathbf{Z}^{-1}y_q$

23:      **end if**

24:      **if** ($k \geq 3$ and $||w - \widehat{w}||_{\infty} \leq C_{tolerance}$) **then**

25:         break

26:      **end if**

27:   **end for**

28:   Finalization

29:   **return** $w$

30: **end procedure**

---

$O(nd\max(d, k))$ when no regularization is applied. In the case without regularization, the time complexity $O(nd\max(d, k))$ is obtained since the computational complexity of the initialization is $O(nd^2)$, and the complexity per iteration is $O(ndk)$. The total computational complexity is $O(nd^2) + O(ndk) \rightarrow O(ndd) + O(ndk)$, which simplifies to

**Table 2 Maximum number of iterations for the scikit-learn logistic regression solvers.**

| Solver | Default maximum iteration ($K$) | Typical iteration count |
|---|---|---|
| LIBLINEAR | 100 | 100 to 1,000 |
| LBFGS | 100 | 100 to 500 (up to 1,000) |
| Newton-CG | | |
| Newton-cholesky | | |
| SAG | 1,000 | 1,000 to 5,000 |
| SAGA | | |

$O(nd \max(d, k))$. For the proposed FBLR method, $k$ is typically less than or equal to the maximum iteration ($K$) with a default value of 10.

Our time complexity analysis reveals that the FBLR method has linear time complexity with respect to the number of rows ($n$) and at most quadratic time complexity with respect to the data dimension ($d$). Furthermore, for the proposed FBLR method, the maximum number of iterations is only 10, which is considerably low. Thus, the maximum iteration count of the proposed FBLR method is small compared to logistic regression in scikit-learn with various solvers (see Table 2), demonstrating its efficiency.

The space complexity of the proposed FBLR method is at least the size of the input data matrix $\mathbf{X} \in \mathbb{R}^{n \times d}$. So, we ignore the vectors employed in the FBLR method, as they have dimensions of $n \times 1$ or $d \times 1$, which are much smaller than $n \times d$, size of the data matrix $\mathbf{X}$. So, we will focus only on the matrices employed in the FBLR method, as outlined below:

- $\mathbf{X}, \mathbf{U}, \mathbf{F}^\top \in \mathbb{R}^{n \times d}$ and $\mathbf{V}, \mathbf{G}, \mathbf{A} \in \mathbb{R}^{d \times d}$ are dense matrices $\rightarrow$ space complexity is $O(nd)$
- $\mathbf{Z}, \mathbf{H} \in \mathbb{R}^{n \times n}$ and $\mathbf{S} \in \mathbb{R}^{d \times d}$ are sparse diagonal matrices $\rightarrow$ space complexity is $O(n)$.

Therefore, space complexity of the proposed FBLR method is $O(nd)$.

## RESULTS

We conducted experiments on a system running Ubuntu Linux 20.04 with an Intel Core i9-10900KF CPU (10 cores) and 64 GB of RAM. Note that NumPy leverages SIMD extensions supported by a CPU to a reasonable extent. For Intel Core i9-10900KF CPU, utilized SIMD extensions by NumPy are SSE, SSE2, SSE3, SSSE3, SSE4.1, SSE4.2, AVX, AVX22, F16C, and FMA3.

For the proposed FBLR method, $\lambda$ and $\gamma$ parameters are set to zero for all experiments unless stated otherwise. Also, fixed default values are used for the parameters $\varepsilon$, $\xi$, $K$, and $C_{\text{tolerance}}$. The parameter $\varepsilon$, a small positive constant for the applied $L_f$-norm approximation, is set to $10^{-10}$. Also, $\xi$, representing the energy percentile for low-rank approximation, is 99.9999, excluding zeros or very small eigenvalues. The maximum number of iterations is set to $K = 10$, and the convergence tolerance is set to $C_{\text{tolerance}} = 10^{-3}$. From this point forward, whenever *logistic regression (LR)* is mentioned in the text, it refers specifically to the *logistic regression* within scikit-learn.

## Datasets

### *Realworld dataset*

We used datasets with binary labels from the OpenML in the experiments. We focused on datasets with no missing data and training times exceeding 3 s when utilizing LR. This strategy focused on selecting datasets that would benefit the most from speed enhancements. Our analysis compared the classification performance and execution efficiency between LR and our FBLR method across different scenarios, such as balanced and imbalanced datasets of medium to large sizes. Additionally, we evaluated the LR and FBLR method on the UCI's HEPMASS dataset, which features rich data from key physics experiments targeting exotic particle discovery and poses a binary classification challenge. Detailed information on the datasets used with their attributes is provided in Table 3. In a binary dataset, the term 'majority' denotes a frequency between 0.5 and 1.0 for the class with more occurrences than the other class. In Table 3, creditcard dataset is imbalanced since it has a high majority value.

### *Synthetic dataset*

The *make_classification* method in the scikit-learn library is a mechanism for creating random synthetic data with a desired number of classes, samples, and features. *n_redundant* is responsible for generating linear combinations of informative features, *weights* sets the sample distribution across classes to introduce imbalance, and *flip_y* alters the class of a specified fraction of samples at random, introducing noise and increasing the complexity of the classification task.

## Experiment design

Using real-world data, three experiments were designed to evaluate classification performance and execution time of LR and FBLR: (a) experiment on HEPMASS data with multiple performance metrics (b) a single-run experiment without regularization on OpenML datasets and (c) an experiment incorporating regularization through various parameters on OpenML datasets. Optimizing hyperparameters is crucial to improve the model's classification performance. We employed the grid search cross-validation (GridSearchCV) method in scikit-learn to find the best models for LR and FBLR, specifically to analyze the impact of regularization on accuracy metrics and execution time.

We also examined the alignment between theoretical computational complexity and practical execution times on synthetic datasets, as these allow for greater control over the data.

## Performance metrics

There exist several performance metrics. Accuracy measures the proportion of true results among all samples, providing an overall view of classification accuracy.

$$\text{Accuracy} = \frac{TP + TN}{FP + FN + TP + TN}$$

where $TP$ is the number of true positives, $TN$ is the number of true negatives, $FP$ is the number of false positives, and $FN$ is the number of false negatives.

**Table 3 Datasets utilized in the experiments along with their attributes.**

| Name | *n* (#samples) | *d* (#features) | Majority% |
|---|---|---|---|
| Kits | 1,000 | 27,648 | 52% |
| Road-safety | 111,762 | 32 | 50% |
| Creditcard | 284,807 | 30 | 99% |
| Airlines | 539,383 | 7 | 55% |
| Colon | 5,100,000 | 62 | 50% |
| HEPMASS | 10,500,000 | 28 | 50% |

Precision measures the proportion of *TP* in all positive predictions, recall measures the proportion of actual positives that are correctly identified, and F1-score is the harmonic mean of precision and recall.

$$\text{Precision} = \frac{TP}{FP + TP} \text{ and Recall} = \frac{TP}{FN + TP} \text{ and F1-score} = \frac{2 \times Precision \times Recall}{Precision + Recall}$$

The area under the curve (AUC) is determined by plotting the true positive rate (TPR) *vs.* the false positive rate (FPR) at various thresholds and calculating the area beneath this curve.

## Impact of solver selection on LR performance

The LogisticRegression class in scikit-learn employs several optimization methods (see Table 1), each tailored for different regularizers, data sizes, and computational needs. However, this leads to differences in execution time and accuracy across the solvers. As seen in Table 4, the execution time of Logistic Regression in scikit-learn varies significantly (from 3.575 to 99.339 s). The Newton-CG and LIBLINEAR solvers demonstrate good accuracy, while the limited-memory Broyden-Fletcher-Goldfarb-Shanno (LFBGS) algorithm, stochastic average gradient (SAG), and stochastic average gradient descent (SAGA) have lower accuracy due to inadequate default iteration limits. It is important to note that in this experiment, we did not use GridSearchCV; instead, each solver was executed individually with its default parameters.

For the LR implementation, each solver given in Table 1 has different computational complexity, some being quite efficient, like LIBLINEAR, but the required number of iterations is much larger than our method (see Table 2). The execution time of the proposed FBLR method remains tightly bounded (from 0.1695 to 0.2721 s, with an average of 0.2482 s) as the penalty (regularization) varies (see Table 5). This demonstrates the efficiency of our minimization scheme, which is relatively unaffected by the choice of regularizer compared to solvers used within the LR (Table 4).

We observed that specific solvers within scikit-learn, like LIBLINEAR, lack CPU-level parallelization support, whereas our approach, leveraging NumPy, enables parallelism through scikit-learn's inherent parallel capabilities. We suggest that scikit-learn developers consider adopting a single NumPy-based solver to simplify maintenance and automatically benefit from NumPy improvements.

**Table 4 Performance of LR solvers on the roadsafety dataset.**

| Solver | Penalty | Time (s) | Accuracy |
|---|---|---|---|
| Newton-CG | $L_2$ | 6.7949 | 0.6949 |
| Newton-CG | No regularization | 5.5673 | 0.6949 |
| LBFGS | $L_2$ | 59.4638 | 0.6211 |
| LBFGS | No regularization | 55.8338 | 0.6290 |
| LIBLINEAR | $L_1$ | 39.889 | 0.6949 |
| LIBLINEAR | $L_2$ | 3.5751 | 0.6935 |
| SAG | $L_2$ | 42.6379 | 0.5819 |
| SAG | No regularization | 42.5621 | 0.5820 |
| SAGA | $L_1$ | 99.0867 | 0.5821 |
| SAGA | $L_2$ | 85.0862 | 0.5821 |
| SAGA | ElasticNet ($\alpha = 0.5$) | 99.3389 | 0.5821 |
| SAGA | No regularization | 85.9067 | 0.5821 |

**Table 5 Performance of FBLR with different penalties for road-safety dataset.**

| $\lambda_{ssr}$ | $f$ | $\gamma$ | Time (s) | Accuracy |
|---|---|---|---|---|
| 0 | 0 | 0.001 | 0.2094 | 0.6951 |
| 0 | 0.5 | 0.001 | 0.2653 | 0.6950 |
| 0 | 1 | 0.001 | 0.2447 | 0.6948 |
| 0 | 1.5 | 0.001 | 0.2380 | 0.6950 |
| 0 | 2 | 0.001 | 0.1695 | 0.6949 |
| 1 | 0 | 0.001 | 0.2558 | 0.6950 |
| 1 | 0.5 | 0.001 | 0.2508 | 0.6948 |
| 1 | 1 | 0.001 | 0.2470 | 0.6948 |
| 1 | 1.5 | 0.001 | 0.2550 | 0.6948 |
| 1 | 2 | 0.001 | 0.2688 | 0.6949 |
| 2 | 0 | 0.001 | 0.2602 | 0.6947 |
| 2 | 0.5 | 0.001 | 0.2690 | 0.6948 |
| 2 | 1 | 0.001 | 0.2721 | 0.6947 |
| 2 | 1.5 | 0.001 | 0.2685 | 0.6947 |
| 2 | 2 | 0.001 | 0.2649 | 0.6947 |

## Experimental results

For the experiments, we used the real-world data in Table 3 and synthetic data created with *make_classification* method in the scikit-learn. Real-world data is divided into 70% training data and 30% test data.

### Experiments on real-world datasets

Table 6 presents the performance metrics, while Fig. 1 shows the receiver operating characteristic (ROC) curves for LR and FBLR with default parameters on the HEPMASS

**Table 6 Multiple performance metrics for LR and FBLR on the HEPMASS dataset.**

| Metric | Train | | Test | |
|---|---|---|---|---|
| | LR | FBLR | LR | FBLR |
| Accuracy | 0.83686 | 0.83583 | 0.83646 | 0.83553 |
| Recall | 0.83588 | 0.83124 | 0.83562 | 0.83104 |
| Precision | 0.83759 | 0.83901 | 0.83708 | 0.83863 |
| F1-Score | 0.83673 | 0.83511 | 0.83635 | 0.83481 |

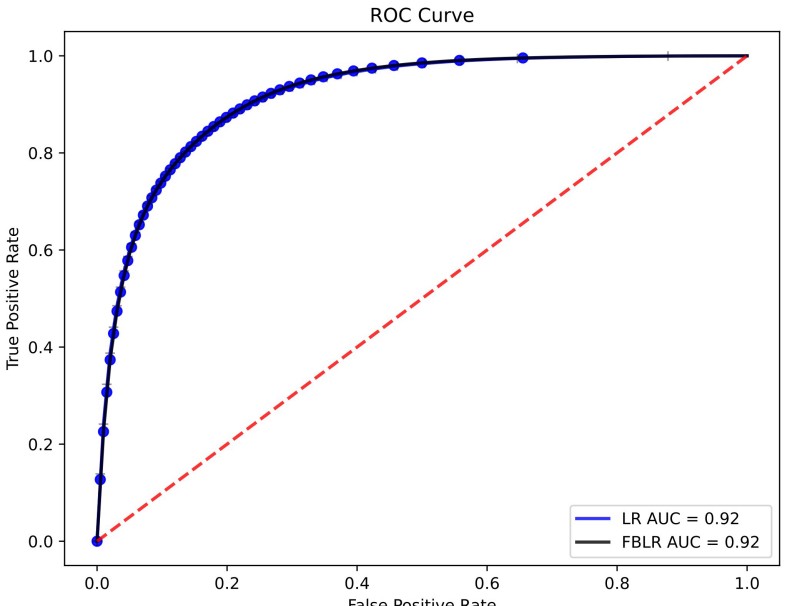

**Figure 1  ROC curves and AUC values of LR and FBLR on HEPMASS dataset.**

dataset. In LR, we used the *LIBLINEAR* solver for faster execution and set the maximum number of iterations to 250 to ensure high accuracy for a fair comparison. We use a single thread, even though FBLR can run with multiple threads, while LR with *LIBLINEAR* cannot. Experiments were run 10 times to compare LR and FBLR, showing that all metrics evaluated had an accuracy loss not exceeding 0.005, while 13.56× speedup is obtained.

LR is powered by *LIBLINEAR*, a highly efficient C/C++ library designed for large-scale linear classification, among other optimized solvers (see Table 1). On the other hand, the proposed FBLR method relies solely on the NumPy package (with openblas64 as the backbone), a core numerical library of Python. This approach ensures that FBLR takes full advantage of NumPy's efficient handling of array operations and integration with the basic linear algebra subprograms (BLAS)[2].

In Table 7, we give the comparison of LR and FBLR for OpenML datasets in terms of accuracy, time (in seconds), and speedup (LR Time/FBLR Time). No regularization is used

[2] BLAS is a collection of low-level routines that accelerate linear algebra computations. BLAS underpins the method's ability to perform computationally intensive tasks more efficiently.

**Table 7 Single-run experiment comparing accuracy and execution time (s) of LR and proposed FBLR method without regularization on OpenML datasets.**

| Filename | LR ACC | FBLR ACC | LR time (s) | FBLR time (s) | Speedup |
|----------|--------|----------|-------------|---------------|---------|
| Kits | 0.5700 | 0.5633 | 4.5378 | 0.6795 | 6.6780 |
| Road-safety | 0.6935 | 0.6956 | 3.7074 | 0.1517 | 24.4337 |
| Creditcard | 0.9991 | 0.9990 | 3.4729 | 0.2766 | 12.5530 |
| Airlines2 | 0.5965 | 0.5939 | 5.0756 | 0.1051 | 48.3072 |
| Colon | 0.9740 | 0.9680 | 10.6571 | 1.0858 | 9.8149 |

**Table 8 GridSearchCV experiment comparing accuracy and execution time (s) of LR and the proposed FBLR method with regularization on OpenML datasets.**

| Filename | LR ACC | FBLR ACC | LR time (s) | FBLR time (s) | Speedup |
|----------|--------|----------|-------------|---------------|---------|
| Kits | – | 0.5667 | – | 18,761.8684 | – |
| Road-safety | 0.6949 | 0.6951 | 191.0991 | 5.8685 | 32.56 |
| Creditcard | 0.9992 | 0.9990 | 49.9478 | 11.1041 | 4.5 |
| Airlines2 | 0.5965 | 0.5956 | 92.7151 | 5.9504 | 15.58 |
| Colon | 0.9740 | 0.9740 | 145.2604 | 77.7610 | 1.87 |

in these experiments. In LR, we set *LIBLINEAR* as a solver parameter and *None* as a penalty parameter are chosen, while for the other parameters, default values are used. Table 7 gives the significant execution time improvements obtained by comparing FBLR to LR. Our method achieves training times up to 48.3 times faster, on average, an order of magnitude faster than LR.

In Table 8, we compare regularized LR and regularized FBLR using GridSearchCV. We used the grid search cross-validation in scikit-learn to find the best models (by searching various parameters) to compare LR and our FBLR method. It is worth mentioning that experimenting with a broader range of parameter combinations could improve both methods' generalization and accuracy. Nonetheless, as our primary objective was to compare the speed of the methods, we maintained an equal number of experiments for consistency. In GridSearchCV, 5-folds are used. It assesses every combination of parameter values to identify the most effective combination, yielding the best classifier. Due to memory limitations in the experimental setup, results for the kits dataset could not be included for the LR algorithm in Table 8 while FBLR is able to process the kits dataset.

Table 8 presents the best accuracies and corresponding execution times and speedups for LR and FBLR on 5 OpenML datasets. For LR, we used the solvers ('lbfgs', 'newton-cg', 'liblinear', 'sag', 'saga') and their penalties ('l1', 'l2', 'elasticnet', None) as GridSearchCV search parameters. As detailed in Table 1, specific solvers may not support certain penalties, so we combine solvers and penalties in GridSearchCV accordingly. To have an equal number of experiments, we limited the FBLR parameters within a specific

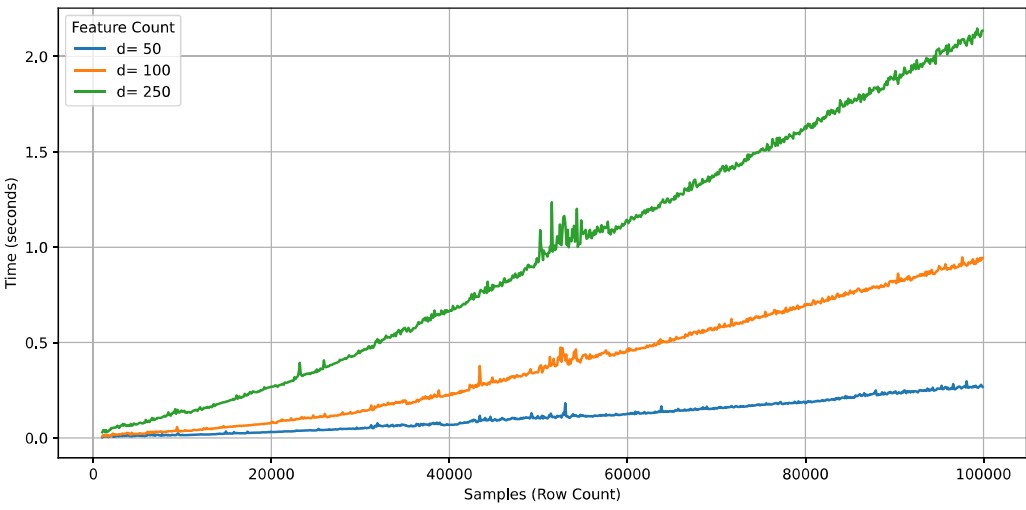

FBLR without regularization using synthetic data

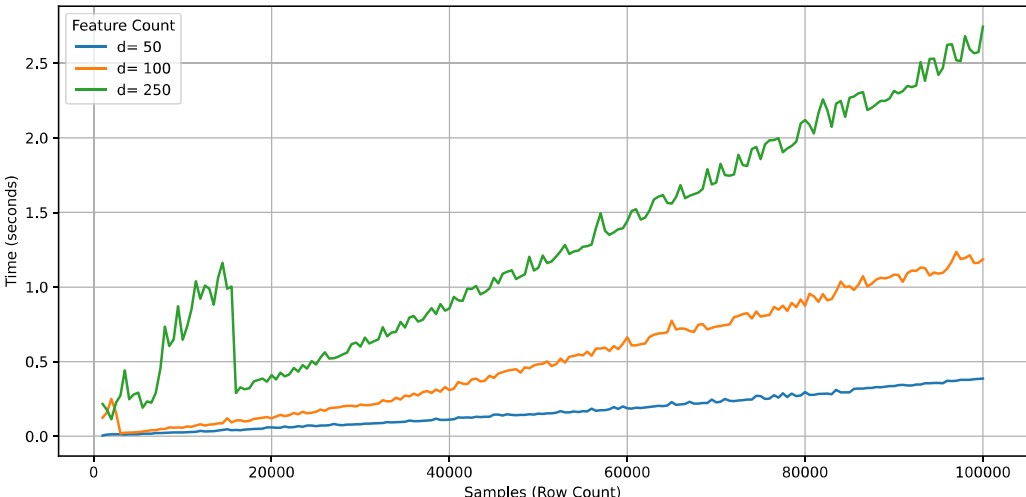

FBLR with $L_f$-norm regularization ($f = 2$, $\lambda = 10^{-10}$, $\gamma = 10^{-10}$) using synthetic data

**Figure 2  Execution time of FBLR with respect to $n$ and $d$ using synthetic data.**

range and tested 12 combinations. In particular, the parameter settings for FBLR in experiments were set as follows: the norm of $L_f$-norm, $f$, was varied over the values [0, 0.5, 1, 2] and the $L_f$-norm regularization coefficient ($\lambda$) over the values [0, 1, 2].

The proposed FBLR method is designed to prioritize computational efficiency while maintaining accuracy. We employ SVD for low-rank approximation, and with a high energy percentile (99.9999), it retains significant information. Our novel second-order Soft-Plus approximation provides a good, though not exact, fit, enhancing numerical minimization efficiency and reducing required iterations ($k$), with minimal potential accuracy loss. If desired, the weight vector from FBLR can be fine-tuned with an efficient Logistic Regression method in just a few iterations for a precise result.

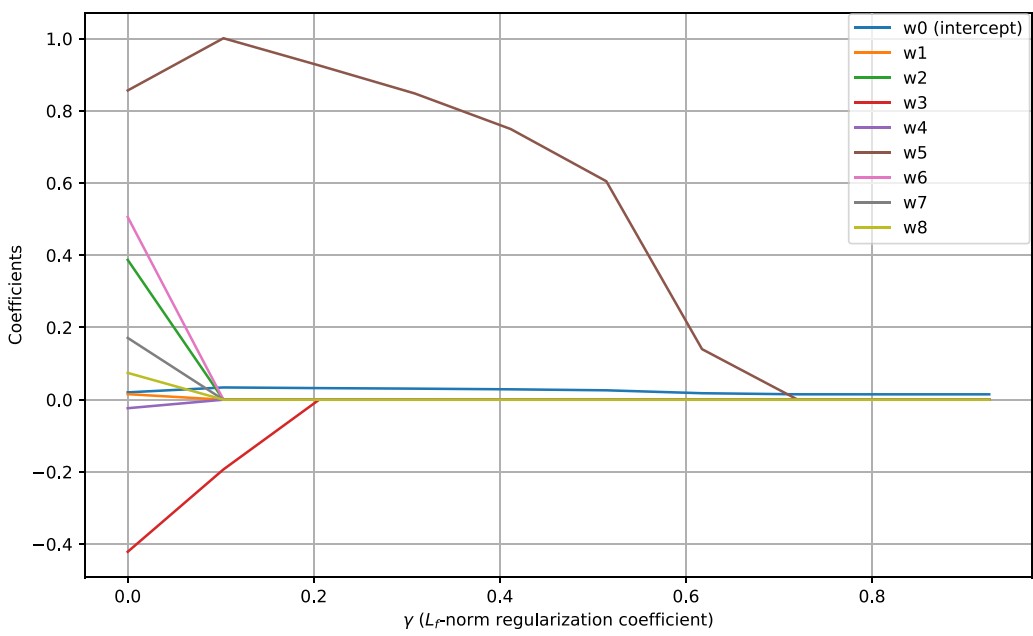

**Figure 3** $L_0$-**norm paths using synthetic data which contains 10K samples with seven features and an intercept.** Feature 2 is a linear combination of feature 0 and 1, and feature 4 is correlated with feature 3.

### Experiments on synthetic dataset

Recall that the computational complexity of the proposed FBLR method is $O(nd^2k)$ when regularization is applied and $O(nd \max(d, k))$ without regularization. Here, $n$ denotes the number of rows, $d$ denotes the number of features, $k$ denotes the number of iterations, and $K$ denotes the maximum number of iterations. When $d \geq K$, the expression $\max(d, k)$ simplifies to $d$, given that $k \leq K$, where $k$ iterates up to a maximum of $K$. Therefore, computational complexity without regularization becomes $O(nd^2k)$. In accordance, Fig. 2 illustrates that the execution time of FBLR scales linearly with $n$ and exhibits a near-quadratic growth with respect to $d$. Note that we analyzed the theoretical and computational complexity with respect to practical execution times on synthetic datasets since we have more control over synthetic data.

Visualization of regularization paths presents change of feature coefficients as the regularization parameter changes. When coefficients of features drop to zero corresponding feature is removed. Figure 3 shows regularization paths for coefficients of the proposed FBLR method under $L_0$-norm regularization. It illustrates the effect of $L_0$-norm regularization in promoting sparsity within the model's parameters ($w_1$ to $w_8$). Sharpness in $L_0$-norm regularization paths and the reduction of weight coefficients to 0 demonstrate the effectiveness of the proposed $L_f$-norm regularizer in promoting sparsity for $f = 0$. Note that, the intercept term, $w_0$, remains unpenalized and almost constant across varying regularization levels determined by the $\gamma$ parameter. Applying regularization without penalizing the intercept is one strong feature of the proposed FBLR

method which is not directly supported by LR. In scikit-learn, non-penalizing the intercept requires non-trivial parameter tweaking and also very data dependent.

## CONCLUSION AND DISCUSSION

This study presents a novel approach, namely fast binary logistic regression (FBLR), to enhance the training efficiency of binary logistic regression. FBLR reduces training times, achieving speeds that are, on average, an order of magnitude faster than those of scikit-learn's logistic regression, courtesy of the efficient numerical minimizer we developed. Moreover, unlike scikit-learn's logistic regression, which utilizes multiple solvers, we propose a single, efficient solver that incorporates a second-order accurate Soft-Plus approximation and flexible $L_f$-norm regularizer. This approach supports various weight penalties, thereby enhancing the training process for versatile models. Compared to scikit-learn, our approach relies on a single solver, simplifying the code and improving maintainability.

Furthermore, we have devised a low-rank approximation strategy to address collinearity among features. Our low-rank approach harnesses randomized SVD for high-dimensional feature sets and also a novel SVD with row reduction (SVD-RR) approach, tailored specifically for large datasets containing numerous rows. For smaller to moderate-sized datasets, we just employ truncated SVD. Also, we introduce an innovative strategy to determine the optimal transition from truncated SVD to randomized SVD. To mitigate the dominance of overly large eigenvalues, the logarithm of the eigenvalues was employed to establish the rank, $r$, suitable for low-rank approximation. The achieved computational efficiency is crucial for crafting generalized logistic regression models, enabling iterative parameter tuning to achieve an optimal balance between bias and variance.

In addition to existing computational efficiencies, further acceleration can be achieved by applying stochastic gradient descent or mini-batch methods for faster approximate gradient evaluation. Also, parallel computing frameworks such as CuPy, Numba, PyCUDA, or PyTorch can be utilized. Additionally, deploying a caching strategy for SVD and storing the compact diagonal sparse matrix $\mathbf{S}$ and the dense square matrix $\mathbf{V}$ in a dictionary can enable efficient reuse. Matrix $\mathbf{U}$ can be computed using Eq. (41) once diagonal matrix $\mathbf{S}$ and square matrix $\mathbf{V}$ are retrieved using the hash code of input data $\mathbf{X}$. This is particularly useful for repeated training sessions when performing best-parameter searches in scikit-learn. Originally conceived for binary labels, our method possesses sufficient flexibility to accommodate multinomial classifications using the inherent functions provided by scikit-learn. Nonetheless, our approach offers no advantage over scikit-learn's logistic regression for handling missing values. Also, similar to scikit-learn's logistic regression, our method is limited to handling linearly separable data. The matrix-vector form of our cost function, resembling Ridge Regression, allows our method to incorporate kernel methods like Kernel Ridge Regression, enabling nonlinear data handling while supporting linear data in its current form. Such kernel method extension can benefit from techniques such as Random Fourier Features (RFF) (*Rahimi & Recht, 2007*) to increase execution time performance further.

# APPENDIX

## APPROXIMATION OF SOFT-PLUS

The quadratic approximation of $\log(1 + e^\tau)$ at $\hat{\tau}$ is defined as:

$$\log(1 + e^\tau) \approx z\tau^2 + q\tau + t \tag{23}$$

where $\hat{\tau}$ is a proxy constant for $\tau$ and $z$, $q$, and $t$ are all constant coefficients.

First, we determine $t$ by substituting 0 to $\tau$ as in Eq. (24):

$$\log(1 + e^0) \approx z0^2 + q0 + t \tag{24}$$
$$t = \log(2). \tag{25}$$

Then, for $\tau = 0$, we have $z = \frac{1}{8}$ and $q = \frac{1}{2}$ using Maclaurin series. Next, for $\tau \neq 0$, to find the value of $q$ and $z$, we compute the equation at $-\hat{\tau}$ and $\hat{\tau}$

$$\log(1 + e^{\hat{\tau}}) \approx z\hat{\tau}^2 + q\hat{\tau} + \log 2 \tag{26}$$
$$\log(1 + e^{-\hat{\tau}}) \approx z\hat{\tau}^2 - q\hat{\tau} + \log 2. \tag{27}$$

By subtracting the Eq. (27) from the Eq. (26) we obtain

$$q = \frac{\log(1 + e^{\hat{\tau}}) - \log(1 + e^{-\hat{\tau}})}{2\hat{\tau}}. \tag{28}$$

Note that,

$$\log(1 + e^{-\hat{\tau}}) = \log(1 + e^{\hat{\tau}}) - \hat{\tau}. \tag{29}$$

Then, we obtain $q = \frac{1}{2}$ by plugging Eq. (29) into Eq. (28).
Finally plugging $q$ and $t$ into Eq. (26), we obtain

$$z = \frac{\log(1 + e^{\hat{\tau}}) - \log 2}{\hat{\tau}^2} - \frac{1}{2\hat{\tau}} \tag{30}$$

which leads to the below approximation at point $\hat{\tau}$:

$$\log(1 + e^\tau) \approx z\tau^2 + \frac{1}{2}\tau + \log 2 \text{ where } z = \begin{cases} \frac{1}{8} & \text{for } \hat{\tau} = 0 \\ \frac{\log(1 + e^{\hat{\tau}}) - \log 2}{\hat{\tau}^2} - \frac{1}{2\hat{\tau}} & \text{otherwise.} \end{cases} \tag{31}$$

Note that our quadratic Soft-Plus approximation is specifically designed for minimizing $-\log l(w)$ and differs from second-order Taylor series-based approximation. Compared to the second-order Taylor approximation, our Soft-Plus approximation more effectively adapts to the shape of the Soft-Plus function, being exact at both the approximation point and the origin. This provides a more accurate quadratic approximation for minimizing the cost function. The second-order Taylor approximation of Eq. (23) at $\hat{\tau}$ is given as below which is different than our approximation:
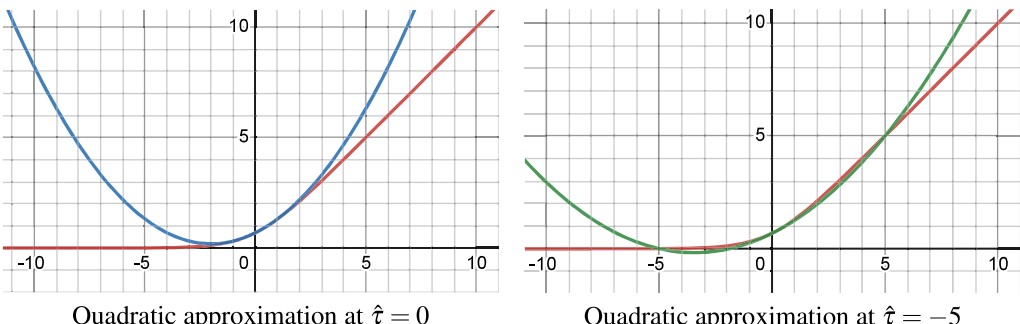

Quadratic approximation at $\hat{\tau} = 0$        Quadratic approximation at $\hat{\tau} = -5$

**Figure 4 Quadratic approximation of Soft-Plus at $\hat{\tau} = 0$ and $\hat{\tau} = -5$.**

$$
\begin{aligned}
\log(1 + e^{\tau}) \approx &\left( \frac{e^{\hat{\tau}}}{2(1 + e^{\hat{\tau}})^2} \right) \tau^2 \\
&+ \left( \frac{e^{\hat{\tau}}}{1 + e^{\hat{\tau}}} - \frac{\hat{\tau} e^{\hat{\tau}}}{(1 + e^{\hat{\tau}})^2} \right) \tau \\
&+ \left( \log(1 + e^{\hat{\tau}}) - \frac{\hat{\tau} e^{\hat{\tau}}}{1 + e^{\hat{\tau}}} + \frac{\hat{\tau}^2 e^{\hat{\tau}}}{2(1 + e^{\hat{\tau}})^2} \right).
\end{aligned}
\tag{32}
$$

**Empirical quadratic approximation of soft-plus**

The employed quadratic approximation of the Soft-Plus function given in Eq. (23) at point $\hat{\tau} = 0$ and $\hat{\tau} = -5$ are shown in Fig. 4. These approximations are exact at both the approximated point ($\hat{\tau}$) and the origin ($\tau = 0$), with high accuracy between them, but become less accurate outside this range. In contrast, the second-order Taylor approximation only guarantees exactness at the approximation point $\hat{\tau}$.

## APPROXIMATION OF $L_f$-NORM

We also approximated smooth $L_f(\rho) = \rho^2(|\rho|^{2-f} + \varepsilon)^{-1}$ quadratically at a point $\hat{\rho}$ where $u$ is defined as a constant. Here, $\hat{\rho}$ is a proxy constant for $\rho$. Quadratic approximation of smooth $L_f(\rho)$ function is:

$$
L_f(\rho) \approx \rho^2(|\hat{\rho}|^{2-f} + \varepsilon)^{-1} = u\rho^2 \text{ where } u = \left( |\hat{\rho}|^{2-f} + \varepsilon \right)^{-1}.
\tag{33}
$$

    Optimization problems with $L_0$ terms are combinatorial, requiring discrete decisions to minimize non-zero elements, making direct approaches computationally impractical for large-scale issues. Smooth approximations are employed as a solution, substituting the $L_0$ with a continuous, differentiable function that mimics the counting of non-zero elements. Similarly, our $L_f$-norm approximation smoothly mirrors the $L_1$-norm for $f = 1$ and $L_0$-norm for $f = 0$, as illustrated in Fig. 5. With our smooth approximation, as $\varepsilon$ gets smaller approximation gets better (default $\varepsilon = 10^{-10}$).

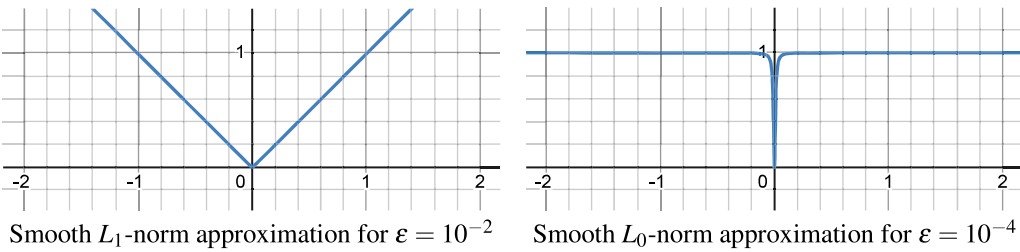

Smooth $L_1$-norm approximation for $\varepsilon = 10^{-2}$     Smooth $L_0$-norm approximation for $\varepsilon = 10^{-4}$

**Figure 5 Smooth approximation of $L_f$-norm for $L_1$-norm and $L_0$-norm.**

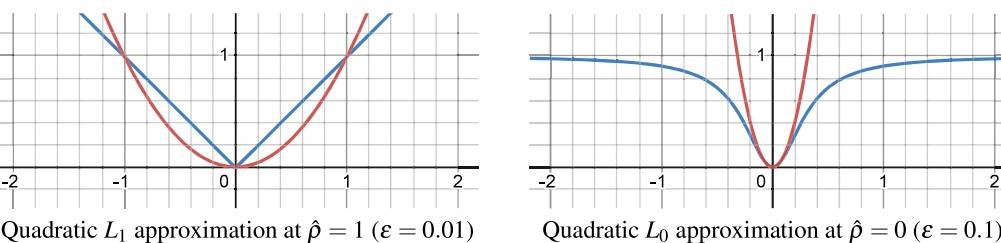

Quadratic $L_1$ approximation at $\hat{\rho} = 1$ ($\varepsilon = 0.01$)     Quadratic $L_0$ approximation at $\hat{\rho} = 0$ ($\varepsilon = 0.1$)

**Figure 6 Quadratic approximation of $L_f$-norm for $L_1$-norm and $L_0$-norm.**

Despite the $L_1$-norm and $L_0$-norm approximation in Fig. 5 being precise even for relatively large $\varepsilon$ values ($\varepsilon = 10^{-2}$ and $\varepsilon = 10^{-4}$), its direct use is unsuitable within our cost function, leading us to employ a quadratic approximation over this smooth approximation. The quadratic approximations of the $L_f$-norm for $L_1$-norm ($f = 1$) and for $L_0$-norm ($f = 0$) are depicted in Fig. 6.

## SVD WITH ROW REDUCTION

In the case of a large data matrix $\mathbf{X} \in \mathbb{R}^{n \times d}$, truncated SVD is computationally demanding due to its complexity $O(nd^2)$. In scenarios where $d^2$ becomes large, the computational burden of applying SVD is mitigated by utilizing randomized SVD (*Halko, Martinsson & Tropp, 2010*; *Halko et al., 2011*; *Martinsson et al., 2010*). However, this method is less effective in dealing with cases where $n$ is large. In such instances, a large subset of the matrix $\mathbf{X}$ can sufficiently mimic the statistical properties of the full data set. Hence, we sub-sample matrix $\mathbf{X}$ to create $\tilde{\mathbf{X}}$, consisting of $m$ rows. This sub-sampling involves randomly or sequentially selecting $m$ rows from $\mathbf{X}$ where we just sub-sampled the first $m$ rows. Note that one can employ a sampling mechanism proposed in *Menon & Elkan (2011)*, such as the Drineas, Kannan, and Mahoney method, to obtain better statistical guarantees for the sampled subset. We use a simple sampling mechanism where we set $m$ to a sufficiently large value, $10^5$, ensuring $\mathbf{X}$'s statistical preservation and reducing computational demands for $n > 2m$, thus outperforming direct SVD on the full matrix. However, $\mathbf{U}$ returned by SVD applied on $\tilde{\mathbf{X}}$ is now $m \times d$ instead of $n \times d$ and also $\mathbf{S}$ requires normalization. To address these issues, we define the covariance matrices for the full matrix $\mathbf{X}$ and the subsampled matrix $\tilde{\mathbf{X}}$ as follows:

$$\mathbf{C} = \frac{1}{n-1}\mathbf{X}^{\top}\mathbf{X} \tag{34}$$

$$\tilde{\mathbf{C}} = \frac{1}{m-1}\tilde{\mathbf{X}}^{\top}\tilde{\mathbf{X}}. \tag{35}$$

Assuming $\tilde{\mathbf{X}}$, a large subset of $\mathbf{X}$, reflects the dataset's statistics, their covariances are nearly equal ($\mathbf{C} \approx \tilde{\mathbf{C}}$):

$$\frac{1}{n-1}\mathbf{X}^{\top}\mathbf{X} \approx \frac{1}{m-1}\tilde{\mathbf{X}}^{\top}\tilde{\mathbf{X}}. \tag{36}$$

Applying the definition of SVD to both matrices, represented as $\mathbf{X} = \mathbf{U}\mathbf{S}\mathbf{V}^{\top}$ and $\tilde{\mathbf{X}} = \tilde{\mathbf{U}}\tilde{\mathbf{S}}\tilde{\mathbf{V}}^{\top}$ respectively, the aforementioned equation can be reformulated as follows:

$$\frac{1}{n-1}(\mathbf{U}\mathbf{S}\mathbf{V}^{\top})^{\top}(\mathbf{U}\mathbf{S}\mathbf{V}^{\top}) \approx \frac{1}{m-1}(\tilde{\mathbf{U}}\tilde{\mathbf{S}}\tilde{\mathbf{V}}^{\top})^{\top}(\tilde{\mathbf{U}}\tilde{\mathbf{S}}\tilde{\mathbf{V}}^{\top}). \tag{37}$$

The above equation can be simplified to:

$$\frac{1}{n-1}(\mathbf{V}\mathbf{S}\mathbf{S}\mathbf{V}) \approx \frac{1}{m-1}(\tilde{\mathbf{V}}\tilde{\mathbf{S}}\tilde{\mathbf{S}}\tilde{\mathbf{V}}) \tag{38}$$

where $\mathbf{U}$ and $\tilde{\mathbf{U}}$ are eliminated since $\mathbf{U}^{\top}\mathbf{U} = \mathbf{I}$ and $\tilde{\mathbf{U}}^{\top}\tilde{\mathbf{U}} = \mathbf{I}$. We assume the same eigenvectors for both data matrices:

$$\mathbf{V} = \tilde{\mathbf{V}}$$

$$\frac{1}{n-1}(\mathbf{V}\mathbf{S}\mathbf{S}\mathbf{V}) \approx \frac{1}{m-1}(\mathbf{V}\tilde{\mathbf{S}}\tilde{\mathbf{S}}\mathbf{V}). \tag{39}$$

Multiplying both sides of the above equation by $\mathbf{V}^{\top}$ from left and right and applying simple algebraic simplifications:

$$\mathbf{S} \approx \sqrt{\frac{n-1}{m-1}}\tilde{\mathbf{S}}. \tag{40}$$

Finally, to compute $\mathbf{U}$, we use SVD of $\mathbf{X}$ given as $\mathbf{X} = \mathbf{U}\mathbf{S}\mathbf{V}^{\top}$, then we multiply both sides by $\mathbf{V}\mathbf{S}^{-1}$, leading to:

$$\mathbf{U} = \mathbf{X}\mathbf{V}\mathbf{S}^{-1}. \tag{41}$$

Thereby, we efficiently compute $\tilde{\mathbf{S}}$ and $\tilde{\mathbf{V}}$ using truncated SVD on the sub-sampled smaller data matrix $\tilde{\mathbf{X}}$ and then we also efficiently compute $\mathbf{U}$ matrix using Eq. (41), $\mathbf{S}$ matrix using Eq. (40), and $\mathbf{V}$ matrix using Eq. (39). Thus, proposed SVD-RR enables computing SVD of a large data matrix $\mathbf{X}$.

## RANDOMIZED SVD

For handling a large matrix $\mathbf{X}$, where truncated SVD may become computationally slow or unfeasible, the utilization of randomized SVD offers a viable alternative, as described in

---

**Algorithm 3** Determine rank using S.

1: **procedure** RANK$\{\mathbf{S}, \xi\}$
2:    $s \leftarrow \log(\text{Diagonal}(\mathbf{S}) + 1)$
3:    $c \leftarrow \frac{\text{CMF}(s)}{\sum_{i=1}^{d} s_i}$                    ▷CMF: Cumulative Mass Function
4:    $r_\xi \leftarrow \text{argmin}_i(c_i > \xi \text{ and } s_i > 10^{-10})$
5:    $r \leftarrow \min(r_\xi, d)$
6:    **return** $r$
7: **end procedure**

---

works by *Martinsson et al. (2010)* and *Halko, Martinsson & Tropp (2010)*, *Halko et al. (2011)*. Randomized SVD have many uses in machine learning with datasets that are too large to be stored in memory (*Halko et al., 2011*). In comparison to classic SVD algorithms, randomized techniques for computing low-rank approximations are frequently faster and, unexpectedly, more robust (*Halko, Martinsson & Tropp, 2010*). Thus, for large datasets where truncated SVD is slow or impractical, so using randomized SVD is suggested by *Martinsson et al. (2010)*.

A critical decision is to determine when to use randomized SVD instead of truncated SVD, taking into account the number of rows, $n$, and the number of features, $d$. To address this issue, we propose to leverage the computational complexities of both approaches as a criterion for decision-making. Computational complexity of the truncated SVD is given as below:

$$O(nd^2). \tag{42}$$

Similarly, computational complexity of the randomized SVD is:

$$O(ndr + nr^2 + r^2 d) \tag{43}$$

where $r$ is the inherent rank of the matrix $\mathbf{X}$. Taking the ratio of randomized and truncated SVD and comparing it with a threshold, we obtained below decision rule (default $T$ is 0.90):

$$\frac{nd^2}{nd\hat{r} + n\hat{r}^2 + \hat{r}^2 d} \leq 2T. \tag{44}$$

However, $\hat{r}$ is unknown until SVD is taken. So, as a practical solution, we estimate $\hat{r}$ using $d$ as shown below:

$$\hat{r} = \min(\sqrt{(d_{limit}d)}, d) \tag{45}$$

where $d_{limit} = 500$ serves as the soft threshold beyond which truncated SVD may begins to exhibit slower performance as $d$ exceeds this value. If $n$ is larger than $10^5$ then $SVD^{(*)}$ computes SVD of $\mathbf{X}$ using SVD-RR ("SVD with Row Reduction"). Otherwise, $SVD^{(*)}$

method gets the **X** as input, uses Eq. (44) to decide to compute truncated SVD or randomized SVD of the data matrix **X**.

## DETERMINING THE RANK

Finally, we developed Algorithm 3 which determines the rank ($r$) using the energy-percentile ($\xi$) and eigenvalues obtained by $SVD^{(*)}$.

### Funding

The authors received no funding for this work.

### Competing Interests

The authors declare that they have no competing interests.

### Author Contributions

- Nurdan Ayse Saran conceived and designed the experiments, performed the experiments, analyzed the data, performed the computation work, prepared figures and/or tables, authored or reviewed drafts of the article, and approved the final draft.
- Fatih Nar conceived and designed the experiments, performed the experiments, analyzed the data, performed the computation work, prepared figures and/or tables, authored or reviewed drafts of the article, and approved the final draft.

### Data Availability

fblr (Fast Binary Logistic Regression) is available at GitHub and Zenodo:

- https://github.com/NurdanS/fblr.

- Nurdan. (2024). NurdanS/fblr: Fast Binary Logistic Regression (fblr) (Version v01). Zenodo. https://doi.org/10.5281/zenodo.14168929.

The HEPMASS database is available at: DOI 10.24432/C5PP5W.

The kits data is available at OpenML: https://www.openml.org/search?type=data&status=active&id=42809&sort=runs.

The road-safety data is available at OpenML: https://www.openml.org/search?type=data&status=active&id=44038&sort=runs.

The creditcard data is available at OpenML: https://www.openml.org/search?type=data&status=active&id=1597&sort=runs.

The airlines data is available at OpenML: https://www.openml.org/search?type=data&status=active&id=1169&sort=runs.

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
