# Peer review of "Fast binary logistic regression"

_PeerJ Computer Science, doi:10.7717/peerj-cs.2579_

## Round 0.1 · original submission · Major Revisions

Dear authors: Please address all the comments from both reviewers, as well as the following feedback from the editor.

Any speedup number such as 48.3 depends on many factors, so do not mention it in the abstract.

Describe the big-O complexity of the algorithms in Table 1 and explain whether the new algorithm is a constant-factor better, or asymptotically better. Mention this in the abstract.

Section 3 must be more clear about which experiments involve hyperparameter search versus which experiments run algorithms from Table 1 just once, for direct comparison with the new algorithm.

Please explain why the new algorithm does not give exactly the same accuracy results as previous algorithms. Most accuracies in Tables 3, 4, 5 are worse for FBLR. Is FBLR faster because it is less accurate?

Table 5 says “road-safety 0.5819 0.6951.” Please explain why these two accuracy numbers are very different.

Explain carefully exactly which version of BLAS is used for the implementation of each algorithm. Note that for Intel CPUs, the Intel Math Kernel Library (oneMKL) is much faster than open-source BLAS versions.

Please do not use the word “performance” because it is ambiguous and can mean “speed” or “accuracy.” Always use one of those words instead (or a different word if “performance” has a third meaning).

·

Basic reporting

This manuscript presents an innovative numerical method aimed at significantly enhancing the training efficiency of binary logistic regression, a widely used statistical model in machine learning. The authors claim that their method achieves an average speedup of up to 48.3 times compared to traditional logistic regression, representing a substantial improvement.

Experimental design

The proposed algorithm is only compared with the classical Logistic regression algorithm, but lacks comparison with other mainstream acceleration algorithms to highlight the advancement of the proposed algorithm.
Logistic regression algorithm can not only solve binary classification problems but also tackle multi-class classification issues. The question arises whether the proposed algorithm can be extended to multi-class classification problems.

Validity of the findings

Novelty and Significance: The proposed Fast Binary Logistic Regression (FBLR) method, with its integration of Lf-norm penalty and advanced singular value decomposition techniques, represents a novel approach to improving logistic regression training efficiency. The claimed speedups are substantial and could have a significant impact on large-scale machine learning applications.
Technical Depth: The authors provide a detailed description of their method, including the use of Lf-norm penalty, singular value decomposition (SVD), and the newly developed SVD-RR technique. This depth of technical explanation is commendable and helps readers understand the intricacies of the proposed approach.
Experimental Evaluation: The authors demonstrate the effectiveness of their method on various datasets from OpenML as well as synthetic datasets. This comprehensive evaluation strengthens their claims about the performance gains.
Weaknesses and Suggestions for Improvement:

Clarity of Motivation: While the introduction provides some context for why improving logistic regression training efficiency is important, a clearer motivation for the specific techniques chosen (e.g., Lf-norm vs. traditional norms, SVD-RR over other dimensionality reduction methods) would strengthen the paper. Clarifying how these choices specifically address the limitations of existing approaches would be beneficial.
Theoretical Analysis: The paper focuses primarily on empirical results. Incorporating some theoretical analysis to explain why and how the proposed techniques lead to the observed speedups would strengthen the technical rigor of the work. This could include bounds on the complexity of the algorithm or convergence guarantees.
Comparison with State-of-the-Art: While the authors compare their method to traditional logistic regression, a comparison with other state-of-the-art approaches for fast logistic regression or similar tasks would provide further insights into the relative strengths and weaknesses of their approach.
Detailed Results and Discussion: Providing more detailed results and a thorough discussion of the findings would make the paper more informative. For example, analyzing the performance gains across different dataset sizes, feature dimensions, and sparsity levels could reveal interesting insights. Similarly, discussing failure cases or limitations of the approach would demonstrate a balanced view.
Clarity of Writing: While the technical content is well-explained, some parts of the writing could benefit from improved clarity. For instance, ensuring consistency in terminology (e.g., using "FBLR" consistently throughout) and revising sentences for readability would enhance the overall presentation.

Additional comments

In summary, this manuscript presents an interesting and promising approach to improving binary logistic regression training efficiency. With some clarifications, additional theoretical analysis, and more comprehensive experimental evaluation, the paper has the potential to make a significant contribution to the field. I recommend the authors consider the above suggestions for improvement and resubmit their manuscript for publication.

Recommendation: Major Revision Required

Cite this review as
Song Y (2025) Peer Review #1 of "Fast binary logistic regression (v0.1)". PeerJ Computer Science

Reviewer 2 ·

Basic reporting

This paper proposes a new optimization loss function for training binary logistic regression models. There are two key components of their approach: (i) they approximate the log(1+exp(tau)) term in the binary logistic regression loss function with a low-order Taylor expansion; (ii) for commonly used regularization terms, they propose to use a fractional norm. Through the epsilon parameter, this fractional norm can approximate L0, L1 and L2 regularization functions.

Experimental design

Experiments on several real and synthetic datasets show that the proposed method achieves on par accuracy with SciKit-Learn's binary logistic regression in a significantly shorter amount of training time. The reported time speedup for training in several scenarios is on average 10x, going up to 48x.

Validity of the findings

Overall, this is a good paper proposing a novel loss to minimize for binary logistic regression, an implementation for fast minimization and providing experimental results to support their hypothesis.

However, the paper can be -- in my opinion, should be -- improved in several ways because the paper is not clear at some parts. I summarize my questions and the issues regarding the paper below.

- In the abstract, authors say "..., which resulted in low ranking." I think there is a language problem here. The sentence must be revised.
- All uses of \cite{} commands are inappropriate. Authors should use \citet{} and \citep{}. They should look up what these commands do. Simply replacing all \cite{}s with \citep{}s would solve most of the problems.
- There is a large paragraph discussing different regularization losses for binary logistic regression in depth. Is this really necessary? Instead, the authors should focus more on how existing methods/implementations try to make training fast.
- In general, table and figure captions are very short. They are not complete. The captions should give sufficient detail so that the reader understand, in which settings the values in the table or plot was obtained.
- What does "none" mean in Table 1? I can understand it but I think it would be better if it is explained.
- The authors criticise scikit-learn's multiple regularization solvers. However, does your method choose the regularizes automatically? Do you have to search for the optimal epsilon value for your method? If yes, then, what is the advantage over scikit-learn? This should be clarified.
- All equations -- even if the text do not refer to them -- should be numbered. For example, as a reader/reviewer, I cannot refer to some of the equations because they don't have numbers.
- How is \hat{w} computed and updated?
- Line145 says "the solution can be represented as below where Z and H are updated in each iteration." However, update equations for Z and H are not provided.
- It is good to show that the proposed method is X times faster than scikit-learn in practice but there is limited theoretical discussion on why this is the case. Is the speedup coming from the Taylor expansion? the fractional norm? why? Can authors comparatively show with math expressions and update equations why the proposed method is faster than a scikit (or liblinear) method?
- In Algorithm 2, it is not clear how Z and H are updated.
- Authors mention grid search and validation several times but there is no information on whether a validation set was used. Please clarify.
- What does "majority" mean in Table 2?
- In Table 3, are the reported values average of 10 runs? Can you also report std dev?
- Line 235: "a speedup factor of 13.56 was observed" -> authors say that there are many variants of LR implementations, yet they do not mention here which variant(s) they are comparing against.
- Line 237-243: To be fair FBLR should be compared against a numpy based LR implementation.
- Line 250: "regularized LR and regularized FBLR" -> which regularizations were used?
- Table 4 caption: caption should be more complete. Which settings did LR and FBLR used?
- Table 4: what is the unit of time?
- What is the purpose of Figure 3? What message does it give to the reader? It is not clear to me.
- Table 5: why is there a variability in speedups? Authors should discuss this.
- Figure 3 caption is not sufficient: which dataset, which settings?

Cite this review as
Anonymous Reviewer (2025) Peer Review #2 of "Fast binary logistic regression (v0.1)". PeerJ Computer Science

---

## Round 0.2 · Minor Revisions

Thank you to the authors for very good and detailed replies to the reviewers.

Given the detailed response, there is no need to send the submission back to the reviewers. The paper can be published after only a few small additional improvements.

A. Make sure all the explanations in the Response are included in the actual paper, in the appropriate places. Make the big-O analysis a separate section. It does not belong under "Implementation."

B. Bring together in one subsection in the paper the warnings to users about the sklearn implementation, and recommendations to maintainers to improve it. If the sklearn defaults often give bad accuracy, the sklearn maintainers should change them. In this subsection, include a version of Table 1 from the Response.

C. Page 4 of the response: "name: openblas64" indicates that MKL is not used. In general, Anaconda and MKL are not open source, so Anaconda makes MKL available. Please double-check which BLAS is used by each algorithm that you measure.

Also pay attention to SIMD extensions. Different CPUs implement different extensions. For good speed, all available SIMD extensions must be used. But for simplicity, libraries often fail to use these.

Consider comparing FBLR to an implementation that is known to have been optimized for speed, namely https://docs.rapids.ai/api/cuml/stable/api/#logistic-regression.

D. Explain in the main paper that your approximation is not just a Taylor expansion, and explain the benefits. This is a significant contribution.

E. There are many different published methods for randomized SVD. See https://dl.acm.org/doi/pdf/10.1145/1921632.1921639 "Fast Algorithms for Approximating the Singular Value Decomposition" (ACM TKDD) which discusses algorithms not mentioned in this submission. Explain in the paper the reasons for your choice of randomized SVD method, and alternatives.

F. "our approach becomes particularly efficient when no regularization is applied" Randomization often provides a type of regularization, such as with dropout for neural networks. Does that happen with FBLR?

---

## Round 0.3 · accepted · Accept

Thank you for addressing the previous feedback thoroughly. The paper can be published now without further reviewing. However, in the final version, the authors should incorporate the following comments.

Line 16 typo: regularize, not regulate.

Tables 6, 7, 8: Liblinear almost always gives higher accuracy. Mention this, explain why, and emphasize fine-tuning (line 375).

Appendix A: Soft-plus is approximately a rediscovery of what is called Lagrange interpolation. A quadratic can guarantee exactness at three points, but the authors only use two. Alternatively, exactness can be guaranteed for fewer points but also for some derivatives; this is Hermite interpolation.

Two papers that have previously used Lagrange interpolation in the context of logistic regression are https://eprint.iacr.org/2018/662.pdf (Efficient Logistic Regression on Large Encrypted Data; see Figure 2) and https://arxiv.org/pdf/2406.13221 (Privacy-Preserving Logistic Regression Training on Large Datasets, Section 4.1).